# Higher Expressions of SHH and AR Are Associated with a Positive Receptor Status and Have Impact on Survival in a Cohort of Croatian Breast Cancer Patients

**DOI:** 10.3390/life12101559

**Published:** 2022-10-07

**Authors:** Ivan Budimir, Čedna Tomasović-Lončarić, Kristina Kralik, Josipa Čonkaš, Domagoj Eljuga, Rado Žic, Božo Gorjanc, Hrvoje Tucaković, Doroteja Caktaš, Josip Jaman, Valentino Lisek, Zlatko Vlajčić, Krešimir Martić, Petar Ozretić

**Affiliations:** 1Department of Plastic, Reconstructive and Aesthetic Surgery, Dubrava University Hospital, School of Medicine, University of Zagreb, 10000 Zagreb, Croatia; 2Faculty of Medicine Osijek, Josip Juraj Strossmayer University of Osijek, 31000 Osijek, Croatia; 3Clinical Department of Pathology and Cytology, Dubrava University Hospital, School of Medicine, University of Zagreb, 10000 Zagreb, Croatia; 4Department of Medical Statistics and Medical Informatics, Faculty of Medicine Osijek, Josip Juraj Strossmayer University of Osijek, 31000 Osijek, Croatia; 5Division of Molecular Medicine, Ruđer Bošković Institute, 10000 Zagreb, Croatia; 6Faculty of Health Sciences, Libertas International University, 10000 Zagreb, Croatia; 7Department of Abdominal Surgery, Dubrava University Hospital, School of Medicine, University of Zagreb, 10000 Zagreb, Croatia

**Keywords:** breast cancer, molecular subtypes, Sonic hedgehog protein, androgen receptor, prognostic biomarkers, survival analysis

## Abstract

Breast cancers (BC) are usually classified into four molecular subtypes according to the expression of estrogen (ER), progesterone (PR), and human epidermal growth factor 2 (HER2) receptors and proliferation marker Ki-67. Despite available anti-hormonal therapies and due to the inherent propensity of some subtypes to develop metastasis, there is a permanent need to discover new prognostic and predictive biomarkers, as well as therapeutic targets for BC. In this study, we used immunohistochemical staining to determine the expression of androgen receptor (AR) and sonic hedgehog protein (SHH), the main ligand of the Hedgehog-GLI (HH-GLI) signaling pathway, in 185 archival primary BC tissue samples and correlated it with clinicopathological characteristics, molecular subtypes, receptors statuses, and survival in a cohort of Croatian BC patients. Results showed that higher SHH and AR expressions were associated with positive receptor status, but increased SHH expression had a negative impact on survival in receptor-negative BCs. On the contrary, higher AR expression was mostly protective. However, multivariate analysis showed that only higher AR expression could be considered as an independent prognostic biomarker for poorer overall survival in triple-negative breast cancer patients (TNBC) (HR 10.9, 95% CI 1.43–83.67; *p* = 0.021), what could be Croatian population-related. SHH could be a potential target for treating TNBCs and HER2-enriched BCs, in cases where HH-GLI signaling is canonical (SHH-dependent).

## 1. Introduction

Breast cancer (BC) is the most commonly diagnosed cancer in all countries, even those with low and intermediate levels of income [1]. The incidence rates are the lowest in Asia and sub-Saharan Africa and the highest in North America, Australia/New Zealand, as well as western and northern Europe [2]. According to the Croatian National Cancer Registry, in 2020 there were 2894 newly diagnosed BC cases (incidence rate 120.3/100,000) and 832 deaths (mortality rate 32.8/100,000) by BC [3]. The societal changes brought by industrialization, such as increased fat intake, body weight, age at menarche and/or lactation, and reproductive patterns such as fewer pregnancies and later ages at first birth, are probably responsible for these global variances [2].

For every newly diagnosed BC, expression of estrogen receptor (ER), progesterone receptor (PR), human epidermal growth factor receptor 2 (HER2) and proliferation marker protein Ki-67 is routinely determined, and according to these characteristics molecular or intrinsic subtypes of BC are defined. A molecular subtype of BC is still among the most important factors in determining the treatment protocol for a patient in most countries. Luminal A (LumA) subtype of BCs expresses ER and PR, while Ki-67 is present in less than 20% of cancer cells. The expression of hormone receptors makes these tumors prone to anti-hormone therapy and considering the low presence of proliferation index, these tumors have better treatment outcome [4]. In the luminal B (LumB) subtype ER and PR are expressed, with PR in usually a lower percentage, but Ki-67 is present in 20% or more cancer cells. This subtype can also overexpress HER2. Even though they are similar to LumA, they are related to worse prognosis due to a higher proliferation index and this subtype presents up to 80% of all BC cases [4]. If tumor cells do not express ER or PR, but do overexpress HER2, the tumor is classified into HER2-enriched (HER2-E) molecular subtype. HER2-E is frequently occurring among younger women, related to lower differentiation and worse treatment outcome [4]. Triple-negative breast cancers (TNBCs) do not express any hormone receptor. They are low differentiated and highly aggressive, with fewer treatment options and a worse prognosis. Women with *BRCA1* mutation most frequently develop this BC subtype [4].

Androgen receptor (AR) is not routinely checked in BC, but several studies have shown that 60–80% of BC patients overexpress AR, and they suggest that AR may be a predictive or prognostic factor as well as a therapeutic target in BC [5,6]. The function of AR appears to vary among the many BC subtypes, and its prognostic and predictive utility in BC patients is still debatable despite numerous publications covering the physiology of AR, AR-related processes, and AR-targeting therapeutics [7,8].

The Hedgehog-GLI (HH-GLI) pathway is a highly conserved signaling pathway from the cell membrane to the cell nucleus. Its role is crucial in stem cell renewal, tissue regeneration, and embryonic development. Core pathway components in mammals are three Hedgehog ligands: desert, Indian, and sonic hedgehog (DHH, IHH, and SHH, respectively); two transmembrane receptors, patched 1 (PTC1) and patched 2 (PTC2); transmembrane protein smoothened (SMO), and three transcriptional factors: GLI1, GLI2, GLI3, which are regulated by suppressor of fused (SUFU) protein [9]. HH ligand is modified in its secretory cell by the addition of a cholesterol and palmitate from palmitic acid into its active state [10]. Binding of HH ligand to PTC stops repression of SMO and that de-repression enables the release of GLIs from SUFU. GLIs translocate to the nucleus and trigger target genes transcription [11]. GLI1 is exclusively a transcriptional promotor while GLI2 and GLI3 can act as both promotors and suppressors. Described activation is considered as a canonical signalization, while non-canonical is an activation of GLI transcription factors independently of SMO [12].

The role of the HH-GLI pathway in breast cancer depends on the histological and molecular characteristics of the cancer, and this topic has been recently extensively reviewed by Riobo-Del Galdo et al. [13]. For instance, significantly higher levels of SHH and DHH were perceived in cancer tissue compared to healthy breast tissue [14]. Higher SHH and DHH expression also positively correlated with cancer invasiveness, lymph node metastases and recurrences [15,16]. In ER-positive (ER+) BCs it was shown that this signaling pathway increases in the proportion of cancer stem cells (CSC) and promotes invasiveness [14,17,18,19]. HER2-positive (HER2+) tumors had higher levels of SHH and GLI1 compared to healthy controls, but these higher levels had no significant correlation with clinical or pathological characteristics [20,21,22]. A study on BC cell lines demonstrated that inhibition of hedgehog acyltransferase HHAT, one of the main enzymes in SHH synthesis, led to a decrease in the growth of HER2+ tumor cells [16]. Several studies confirmed that higher expression of SHH and SMO in TNBC tumors positively correlates with higher nuclear grade, while higher disease stages correlated with SMO and GLI expression [14,15]. A high proportion of CSC was also observed, followed by a higher incidence of recurrences and metastases [23,24]. High SHH expression in in vitro conditions positively correlated with cancer cell proliferation, also canonical signaling promoted metastases development through neo-angiogenesis promotion [25,26]. Research on animals showed a predisposition for the development of bone metastases in TNBC tumors with high SHH levels [27,28].

Recently it was shown by Trnski et al. that in androgen-independent prostate cancer (PC) cells SHH binds to AR through its cholesterol modification and activates AR signaling, which sustained androgen independence [29]. Sabol et al. in their study gave an indication of potential direct interaction between SHH and ERα in the MCF-7 BC cell line [30]. We hypothesize that SHH can bind to sex hormone receptors with its cholesterol modification but before any mechanistical studies, first we wanted to determine the expression of AR and SHH in breast cancer tissue samples from the Croatian population and examine potential correlations between this expression and clinical and pathological characteristics, especially BC molecular subtypes and receptors statuses. We also wanted to assess the prognostic significance of SHH and AR expressions, which could bring those two proteins closer to clinical practice. To the best of our knowledge, this is the first study focusing on both SHH and AR in BC.

## 2. Materials and Methods

### 2.1. Archival Tumors Tissue Samples and Patients Data Collection

This study was conducted on 185 archival primary BC tissue samples collected from the Clinical Department of Pathology and Cytology, Dubrava University Hospital, Zagreb. Samples were obtained from women non-consecutively operated between 2010 and 2015 at the Department of Plastic, Reconstructive and Aesthetic Surgery at the same hospital. The following clinical and pathological data were collected from present medical documentation: date of diagnosis, patient’s age, tumor histotype, T, N, M status, the status of ER, PR and HER2 receptors, level of proliferation marker Ki-67, and presence of lymphovascular invasion, local and distant disease recurrence. Data on survival were collected from the Hospital Information System (BIS) and by phone calls to patients. The study was conducted according to the guidelines of the Declaration of Helsinki and approved by the Ethics Committee of the Dubrava University Hospital (protocol code 2020/2409-07; approved 24 September 2020). Description of patients and tumor samples included in this study is presented in Table 1.

### 2.2. Determination of Androgen Receptor Protein Expression

To determine AR protein expression, 2–3 µm thick sections were cut from archival paraffin blocks, followed by deparaffinization in thermostat. Deparaffinization was followed by pre-digestion in a thermal bath PT Link (DAKO, Glostrup, Denmark) with EnVision FLEX, High pH (K800021, DAKO) target retrieval solution. Immunohistochemical (IHC) staining was performed using automated DAKO Autostainer Link 48 (DAKO) system, with monoclonal mouse anti-human AR primary antibody, clone AR441 (M356201, DAKO), 1:75 dilution. Immunostains were detected by an indirect avidin-biotin complex technique using REAL EnVision Detection System (K5007, DAKO), followed by contrasting with hematoxylin (1 min), rinsing in an ascending series of alcohol (70–100%) and xylene, and applying a coverslip. Human breast tissue embedded in paraffin was used as a positive control. All prepared slides were microscopically analyzed with an Olympus BX53 microscope (Olympus, Tokyo, Japan) by an experienced pathologist. Androgen receptor positivity for each tissue sample was expressed as the percentage of stained tumor nuclei.

### 2.3. Determination of Sonic Hedgehog Protein Expression

To determine SHH protein expression, 2–3 µm thick sections were cut from archival paraffin blocks, followed by deparaffinization in thermostat, and pre-digestion in Ventana BenchMark Ultra (Roche Diagnostics, Basel, Switzerland) instrument with thermal stands and ULTRA Cell Conditioning Solution (950–224, Roche Diagnostics, Basel, Switzerland). IHC staining was performed using Ventana BenchMark automated immunohistochemical system (Roche Diagnostics), with polyclonal rabbit anti-SHH (H-160) primary antibody (sc-9024, Santa Cruz Biotechnology, Dallas, TX, USA), 1:50 dilution. The ultraView Universal DAB Detection Kit (760–500, Roche Diagnostics) was used for immunostaining visualization, followed by contrasting with hematoxylin (1 min), rinsing in an ascending series of alcohol (70–100%) and xylene, and applying a coverslip. As a positive control, ureteral tissue embedded in paraffin was used. All prepared slides were analyzed with an Olympus BX53 microscope (Olympus) by an experienced pathologist.

For SHH expression, both staining intensity and percentage of stained tumor cells were evaluated. Staining intensity was expressed numerically as 0 (negative), 1 (weak), 2 (medium), and 3 (strong intensity). If any sample showed different staining intensities, first was determined which part of the tumor (percentage of tumor cells) was stained with a stronger intensity, while the remaining part of the tumor tissue was stained with a weaker intensity. The total expression of SHH was expressed as an immunoreactive score (IRS), which is the sum of the product of the stronger intensity with the percentage of cells with the stronger intensity and the product of the weaker intensity with the percentage of cells with the weaker intensity.

### 2.4. Statistical Analysis

The normality of distribution of continuous variables was tested with the Shapiro–Wilk test. Cut-off values used to dichotomize protein expression into ‘low’ and ‘high’ categories were determined using the area under the receiver operator characteristic curve (AUC-ROC) analysis. The chi-square test and Fisher’s exact test were used to assess the association and distribution of categorical variables. Categorical variables were presented with number of samples and percentage. Survival curves were calculated with the Kaplan-Meier method and compared by the log-rank test. The Cox proportional hazards regression model with a forward stepwise variable selection was used for multivariate analysis. Statistical analysis was performed using MedCalc Statistical Software v20.114 (MedCalc Software bv, Ostend, Belgium). A two-tailed *p*-value less than 0.05 was considered statistically significant.

## 3. Results

### 3.1. Association of Clinicopathological Characteristics with Molecular Subtype and Receptor Status in a Cohort of Breast Cancer Patients from Croatia

In this study, a cohort of 185 breast cancer patients operated in a single Croatian institution was included to infer the association of clinicopathological characteristics with molecular subtype and receptor status. Results of the statistical analysis are presented in Appendix A. It could be seen that there was a trend for TNBC patients to be younger at diagnosis, but age was not related to the status of any of the studied receptors. The coexistence of ductal carcinoma in situ (DCIS) in invasive ductal breast cancer (IDC) was significantly more frequent in sex hormone receptor-positive patients, or more precisely, exclusively in luminal A and luminal B subtypes (*p* = 0.0002). TNBC patients had the highest proportion of tumors larger than 2 cm (*p* = 0.0003), which was also observed in ER- (*p* = 0.0004) and PR- (*p* = 0.019) patients, and conversely, in HER2+ (*p* = 0.003) patients. HER2+ patients also had a larger number of positive lymph nodes compared to HER2- (*p* = 0.002). Significantly higher Ki-67 positivity was observed in patients with ER- (*p* < 0.0001), PR- (*p* = 0.0005) or HER2+ (*p* = 0.0003) receptor status, the same as for larger tumors. Lymphovascular invasion (LVI) was most often present in patients with luminal B BC (*p* = 0.029). Death (by any cause) was not significantly more frequent in any of the subtypes but was marginally more frequent in ER- patients. In the same subcohort of patients, there were significantly more cases of disease recurrence (*p* = 0.014). Development of distant metastasis was most frequent in TNBC patients (*p* = 0.002), and it was associated with ER- (*p* = 0.001) and PR- (*p* = 0.004) status.

### 3.2. Expression of Sonic Hedgehog Protein and Androgen Receptor and Its Association with a Molecular Subtype and Receptor Status of Breast Cancer Patients from Croatia

AR and SHH protein expression was determined immunohistochemically on 185 BC formalin-fixed, paraffin-embedded tissue samples. AR was expressed in 91.4% (169/285) of samples. Median AR positivity was 80%, interquartile range 50–90%. Figure 1 and Appendix A show different percentages of AR+ tumor nuclei.

SHH was expressed in 98.4% (182/18) of tissue samples. Median SHH IRS was 150, interquartile range 100–190. Figure 2 and Appendix A show different levels of SHH protein expression.

Next, we tried to associate AR and SHH protein expression with clinicopathological characteristics and receptor status of BC patients. Results are presented in Table 2. In general, AR expression was significantly associated with more characteristics than SHH expression. AR expression was higher in older patients (*p* = 0.017) and those with the coexistence of DCIS in IDC (*p* = 0.017), and that was also the trend for higher SHH expression. Both higher AR and SHH expression were observed in smaller tumors (*p* = 0.002 and *p* = 0.026, respectively), while higher AR was also observed in patients with lower Ki-67 (*p* = 0.001). There was also a trend for higher both AR and SHH expression in patients with a higher number of positive lymph nodes. AR expression was highest in LumA, and lowest in TNBC patients. Later was also observed for SHH expression, while SHH was highest in HER2-E patients. This was also observed in relation to the status of each studied receptor: AR expression was significantly higher in ER+ (*p* < 0.0001), PR+ (*p* < 0.0001), and HER2+ (*p* = 0.008) patients, while SHH was significantly higher in ER+ (*p* = 0.041) and HER2+ (*p* = 0.028), and a similar trend was observed in PR+ patients. Both disease recurrence and distant metastases were significantly more frequent in patients with lower AR expression (*p* = 0.003 and *p* = 0.020, respectively), while disease recurrence was marginally associated with higher SHH expression.

We also wanted to explore if there was any relation between SHH and AR expression, however, correlation analysis showed there was no statistically significant correlation between SHH and AR expression in any subgroup of BC patients (Table 3).

### 3.3. Impact of Clinicopathological Characteristics and Sonic Hedgehog Protein and Androgen Receptor Expressions on Survival of Croatian Patients with Breast Cancer

Furthermore, we wanted to assess the prognostic significance of SHH and AR expression in BC patients. Survival data were collected for all 185 patients. The median follow-up time was 82 months, range 2 to 126 months. Overall survival (OS) was defined as the time from diagnosis to the death from any cause or last check-up, recurrence-free survival (RFS) or time to recurrence was defined as the time from diagnosis to the local or regional disease recurrence or last check-up, while metastasis-free survival (MFS) or time to metastasis was defined the time from diagnosis to the appearance of distant metastasis or last check-up.

Kaplan-Meier survival analysis was performed on the whole cohort of patients as well as separately in each of the molecular subtypes and according to the receptors’ status. Results are presented in Appendix A and Table 4. Generally, higher T and M stage showed a negative impact on OS and MFS in the complete cohort as well as in TNBC, ER- and PR- patients, while histotype did not show any significant impact. Higher SHH expression showed negative impact on OS in HER2-E (*p* = 0.043) and ER- (*p* = 0.044) patients, and negative impact on MFS in TNBC (*p* = 0.042) and PR- (*p* = 0.044) patients (Figure 3 and Figure 4). However, higher SHH expression showed to be associated with a longer time before developing metastasis in ER+ (*p* = 0.044) patients. Higher AR expression showed negative impact on OS in TNBC (*p* = 0.006), ER- (*p* = 0.022) and PR- (*p* = 0.040) patients, while positive impact on OS in HER2+ (*p* = 0.012) patients. Higher AR expression also showed positive impact on MFS in the complete cohort (*p* = 0.018), TNBC (*p* = 0.008), ER- (*p* = 0.024), PR- (*p* = 0.044) and HER2+ (*p* = 0.034) patients. Interestingly, RFS was mostly associated with AR or SHH expression, and not clinicopathological characteristics. For instance, higher SHH expression was related to shorter RFS in the complete cohort (*p* = 0.022), TNBC (*p* = 0.024), PR- (*p* = 0.024) and HER2- (*p* = 0.012) patients, while to the contrary, higher AR expression was positive prognostic biomarker for longer RFS in the complete cohort (*p* = 0.001), ER+ (*p* = 0.045), PR- (*p* = 0.029), PR+ (*p* = 0.042) and HER2- (*p* = 0.015) patients (Figure 3 and Figure 4).

Finally, to infer if either SHH or AR could be considered as an independent prognostic biomarker, the Cox regression analysis was performed. We analyzed only those combinations of endpoints and subgroups of BC patients in which, besides SHH or AR expression, at least one clinicopathological characteristic showed a statistically significant impact on survival rate (Table 4). Results are presented in Appendix A. Only higher AR expression showed to be an independent prognostic biomarker for shorter OS in TNBC patients (hazard ratio 10.9, 95% confidence interval 1.43–83.67; *p* = 0.021).

## 4. Discussion

Newly diagnosed breast cancer is most commonly classified into one of several molecular or intrinsic subtypes according to the status or level of ER and PR sex hormone receptor expression, growth factor receptor HER2, and proliferation marker Ki-67. These subtypes present distinct biological features resulting in different treatment approaches and clinical outcomes [31]. BC-related deaths are caused by the development of metastases, which is inherent to TNBC and HER2-E subtypes, but also induced by a development of therapy resistance [32,33]. Therefore, there is a permanent need to discover new and better prognostic and predictive biomarkers, as well as new targeted or combination therapies [34]. One potential source of such biomarkers and therapeutical targets is the HH-GLI pathway, a developmental signaling pathway whose role in BC etiopathology and significant overexpression of pathway’s members in BC tissues compared to healthy controls have been confirmed in a multitude of studies (reviewed in [13]). However, an association of the expression of the HH-GLI pathway members with BC subtypes, clinicopathological characteristics and prognosis are still inconsistent and often contradictory. This is usually explained by differences in study populations and sample sizes, methodologies used for expression determination, approaches used for quantifying the level of expression and cut-off values used to dichotomize expression into ‘low’ and ‘high’ categories. Instigated by the recent discovery that SHH, the main ligand and activator of signal transduction through the HH-GLI pathway, can physically bind to AR in androgen-independent PC cell lines [29] and that there are indications for potential physical interaction between SHH and ERα [30], in this study we wanted to determine the expression of AR and SHH in breast cancer tissue samples and correlate it with clinicopathological characteristics, receptors statuses and survival in a cohort of BC patients from Croatia.

In our study, SHH was expressed in the vast majority of BC tissue samples. We found a negative correlation between SHH expression and tumor size, which is opposite to previous studies [21,35,36]. This could be explained by a disproportionately smaller number of T3 tumors in our cohort or the fact that studies that have shown a positive correlation between SHH expression and tumor size primarily studied SHH mRNA expression. However, when we correlated SHH expression, dichotomized according to the median value, with three tumor size groups, the highest expression was observed for T3, then for T1, and the lowest for the T2 group (data not presented). But since we had only nine T3 samples, we decided that merging T2 and T3 groups would give more statistically reliable results. SHH was the most highly expressed in HER2-E tumors, less in luminal, and least in TNBC. When we further subdivided our cohort according to the status of each receptor, we observed that SHH expression was significantly higher in ER+ and HER2+ subgroups, and this is consistent with previous findings [15,36,37,38]. So far, the role of this pathway has been mostly studied in hormone receptor-positive and TNBCs, while there is a very small number of published studies describing crosstalk between HER2 and HH-GLI pathways. However, it is known that HER2 activates that pathway through the PI3K/AKT/mTOR signaling in human esophageal adenocarcinoma [39]. An aspirant regulator of SHH expression in BC is the transcription factor nuclear factor NF-kappa-B [40,41,42]. NF-kappa-B binds to the SHH promoter when it is demethylated, while it is known that levels of methylation regulate SHH expression [43]. Also, hypomethylation of SHH promoter was observed in BCs with increased levels of SHH and nuclear staining of NF-kappa-B [20]. In ER-positive BCs increased SHH expression is caused by estrogen, while it is known that estrogen also increases levels of GLI transcription factors [14,18]. In TNBC, activation of the HH-GLI pathway can be both canonical and non-canonical, while paracrine secretion of SHH by TNBC cells affects tumor stroma which contributes to tumor growth, invasion, and metastasis [21,44]. In our study, we observed a marginal positive association of SHH expression with nodal involvement which was observed in several previous studies [36,45]. We did not confirm the previously observed association of increased SHH expression with local [19,45,46] and distant [15,21,36] disease relapses, but in our cohort, there were only eight cases of local and 19 of distant relapses. Despite that, and although there was a relatively small number of patients who died during the follow-up period (24%), we observed a significant negative impact of increased SHH expression on different types of survival in our cohort of BC patients, which has been consistently reported previously [19,21,36,45,47,48]. However, in our study SHH expression has not proven either its significance or ‘independence’ from established BC prognostic biomarkers in the Cox regression analyses, which could be easily addressed to the small number of patients with a certain characteristic who reached studied endpoints. Nevertheless, one of the most interesting findings of our study is the observation that although an increased SHH expression was associated with a receptor-positive status, an increased expression mostly had a negative impact on the survival of sex hormone receptor-negative and TNBC patients, while the opposite was observed for HER2-E patients in whom a higher SHH expression was associated with both HER2+ status and worse prognosis. This was additionally proven by the observation that higher SHH expression showed a positive impact on longer MFS only in ER+ patients. These findings indicate a differential impact or significance of commonly observed increased SHH expression (or activated HH-GLI signaling pathway in general) in BCs, further distinguishing its protective or harmful impact regarding the receptor’s status. The present body of evidence could point to a hypothesis that the activated HH-GLI pathway is protective for sex hormone receptor-positive BCs, but harmful for TNBC and HER2-E BC patients. If additionally proven, this could lead to the application of SHH-inhibitors for the treatment of triple-negative, or hormone receptor-negative in general, and HER2-E BCs [49]. There are several chemical approaches on diminishing SHH activity. For instance, the SE1 monoclonal antibody prevents the binding of SHH to PTC1 [50], but the effectiveness of this and other anti-SHH antibodies against BC has so far been proven only preclinically [21,51,52]. Other inhibitors of SHH/PTC1 interaction are a small molecule robotnikinin [53], pyrimidine derivative 7_3d3 [54], and macrocyclic peptide inhibitor HL2-m5 [55]. On the other side, the final step in posttranslational processing of SHH is catalyzed by hedgehog acyltransferase HHAT so its inhibitors such as RU-SKI 43 can affect SHH processing and thus its activity as a ligand [56]. It was observed that RU-SKI 43 reduces proliferation and anchorage-independent growth of BC cells [22]. Here we must emphasize that all those approaches would not be effective if the HH-GLI pathway is activated in a non-canonical, ligand-independent manner, e.g., by crosstalk with other signaling pathways [57], and also that in the same subtype of BC both types of pathway activation could be present. In those cases, either inhibitors of downstream signaling, such as GLI-inhibitor GANT61 [58], or combination therapy would be more useful [59]. However, although there have been several ongoing clinical trials (reviewed in [13]), no inhibitors of the HH-GLI pathway, either as monotherapy or in combination, are being used in common clinical practice to treat BC.

The importance of AR in BC is being increasingly emphasized in scientific literature and although it is still not determined routinely in clinics, according to some studies, it is overexpressed in 60–80% of BCs, which is why it could have therapeutic and prognostic importance [60]. Most studies attributed a protective effect of AR on BC, although there is still no consensus on this [61,62,63]. On the contrary, quadruple negative breast cancers (QNBCs), so those that lack in ER, PR, HER2, and AR expressions, are considered more aggressive with the worst prognosis [64]. In our study, we find positive AR nuclear staining in more than 90% of samples. Higher AR expression was associated with higher age, the coexistence of DCIS in IDC, smaller tumors, lower Ki-67, luminal and hormone receptor-positive BCs, and fewer local and distant disease relapses, which is all consistent with previous studies and also confirms the protective effect of AR in BC [8,60,62,63]. AR expression was not present in 9% of our samples, all belonging to the TNBC subtype, but this subgroup, which can be considered as QNBC, did not show any differences compared to TNBC cases with AR expression (data not shown). AR was expressed in 66% of our TNBC cases and this subtype generally showed the lowest AR expression, which is consistent with previous studies [60,61,65,66]. Interestingly, higher AR expression showed to be a marker for shorter OS in receptor-negative and TNBCs, of which multivariate Cox regression analyses showed that AR expression could be considered as an independent biomarker for poorer OS only for TNBCs, which would contradict generally considered protective role of AR in BC. However, one multi-institutional meta-analysis showed that the protective or harmful effect of AR on the prognosis of TNBC patients could be population-specific, namely, Bhattarai et al. observed that AR-positive status was a marker of good prognosis in US and Nigerian cohorts, a marker of poor prognosis in Norwegian, Icelandic and Indian cohorts, and neutral in UK cohort [66]. This is a very interesting observation that needs confirmation in a larger population of TNBC patients from Croatia. However, our findings that higher AR expression is protective for RFS and MFS could indicate that this population-specific observation could be not BC-related.

The main reason why we were also interested in AR expression in BC was the idea that physical interaction between SHH and AR observed in PC cell lines could be also present in BCs that express AR. However, our correlation analyses show that there was no correlation between the expressions of those two proteins in any of the subgroups of BC patients related to clinicopathological characteristic, molecular subtype, or receptor status, although the correlation between SHH and AR expression was previously observed in PC tissue samples [67]. We must be aware that physical interaction between those two proteins has been proven in androgen-independent PC cells, which is not a native condition for the PC cell line used in that study [29]. Since we analyzed only primary, treatment-naïve BCs, maybe different results would be observed if we analyze local or distant metastases, or treatment-resistant tissue samples.

We are aware of the limits of our study. Our cohort of Croatian BC patients is relatively small and may be non-representative since all patients are coming from a single center. A small number of patients in some of the subgroups or those who have reached studied endpoints may have influenced observed results. Most observed *p*-values were relatively high and would not remain significant after an adjustment for multiple comparisons. Nevertheless, observed results have encouraged us to continue with research on the role and prognostic significance of the HH-GLI pathway in different molecular subtypes of BCs, and that the possible physical interaction of SHH with sex hormone receptors could be a potential therapeutic target for BC.

## 5. Conclusions

Expression of SHH protein in the Croatian cohort of breast cancer patients was associated with a molecular subtype or more precisely, higher SHH expression was observed in receptor-positive BCs. Conversely, the negative impact of increased SHH expression on survival was observed in receptor-negative BCs, indicating a distinct impact regarding the receptors’ statuses. Increased expression of androgen receptor was associated with clinicopathological characteristics which are associated with less malignant BCs, thus proving its protective role in BC. However, the observation that higher AR expression is an independent prognostic biomarker for worse overall survival of TNBC patients could be Croatian-specific. Additional studies are needed to dis/prove the prognostic and predictive significance of interactions between the Hedgehog-GLI signaling pathway and hormone receptors in BC.

## Figures and Tables

**Figure 1 life-12-01559-f001:**
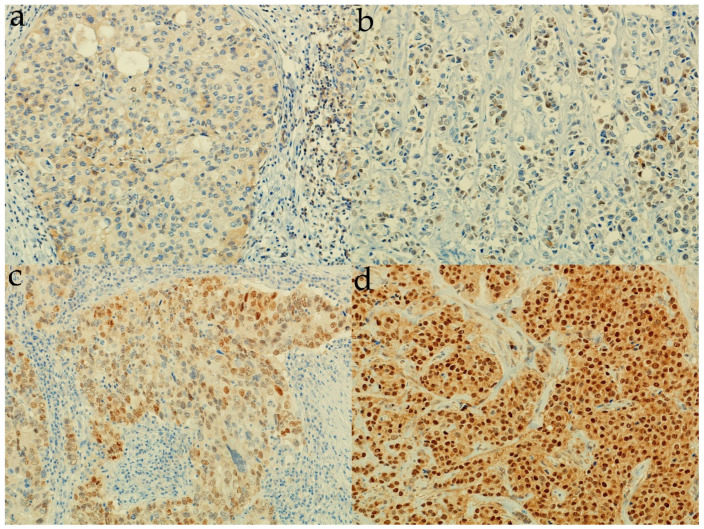
Representative images of tissue samples with different percentages of tumor nuclei with androgen receptor protein expression. (**a**) 0% of AR+ tumor nuclei (200× magnification); (**b**) 30% of AR+ tumor nuclei (200×); (**c**) 60% of AR+ tumor nuclei (200×); (**d**) 100% of AR+ tumor nuclei (200×).

**Figure 2 life-12-01559-f002:**
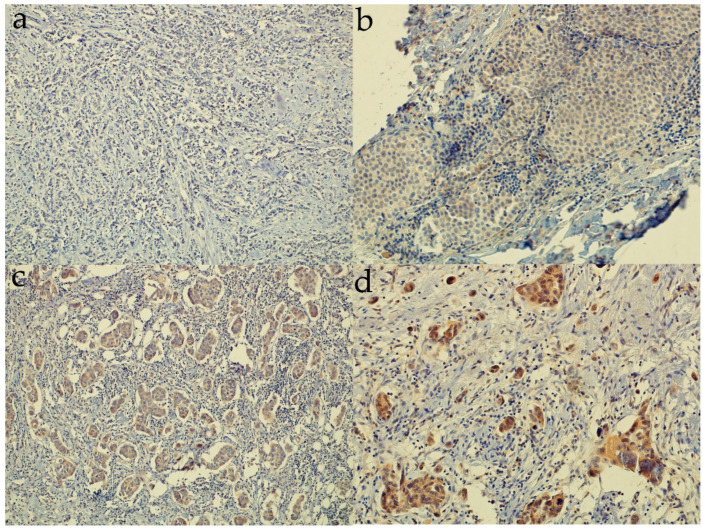
Representative images of tissue samples with different intensities of sonic hedgehog protein expression, expressed as immunoreactive score (IRS). (**a**) IRS 0 (100× magnification); (**b**) IRS 100 (200×); (**c**) IRS 200 (200×); (**d**) IRS 300 (200×).

**Figure 3 life-12-01559-f003:**
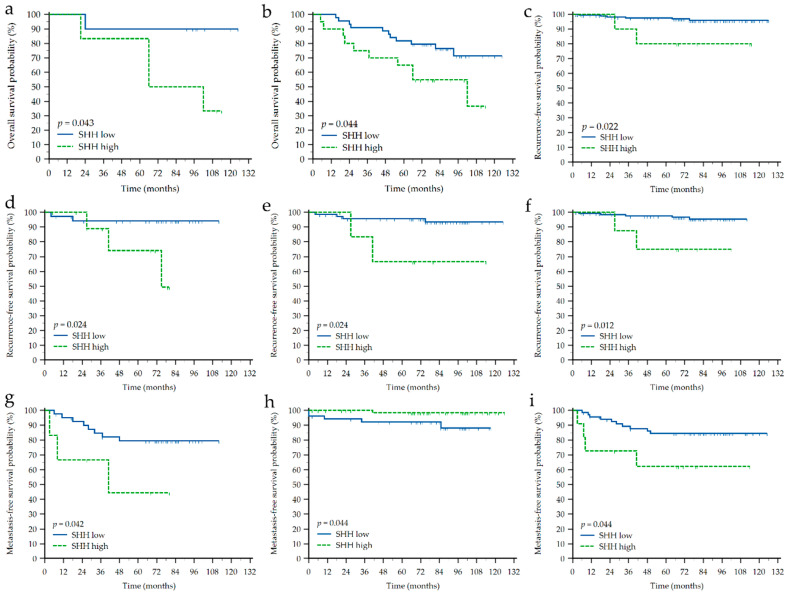
Kaplan-Meier survival curves depicting significant impact of sonic hedgehog (SHH) protein expression on different types of survival in different groups of breast cancer patients. (**a**) Impact on overall survival (OS) in HER2-enriched (*n* = 16); (**b**) OS in ER-negative (*n* = 64); (**c**) Impact on recurrence-free survival (RFS) in the whole cohort (N = 185); (**d**) RFS in TNBC (*n* = 47); (**e**) RFS in PR-negative (*n* = 79); (**f**) RFS in HER2-negative (*n* = 142); (**g**) Impact on metastasis-free survival (MFS) in TNBC (*n* = 47); (**h**) MFS in ER-positive (*n* = 121); (**i**) MFS in PR-negative (*n* = 79). Shown are *p*-values for the long-rank test. Tick marks indicate censored cases.

**Figure 4 life-12-01559-f004:**
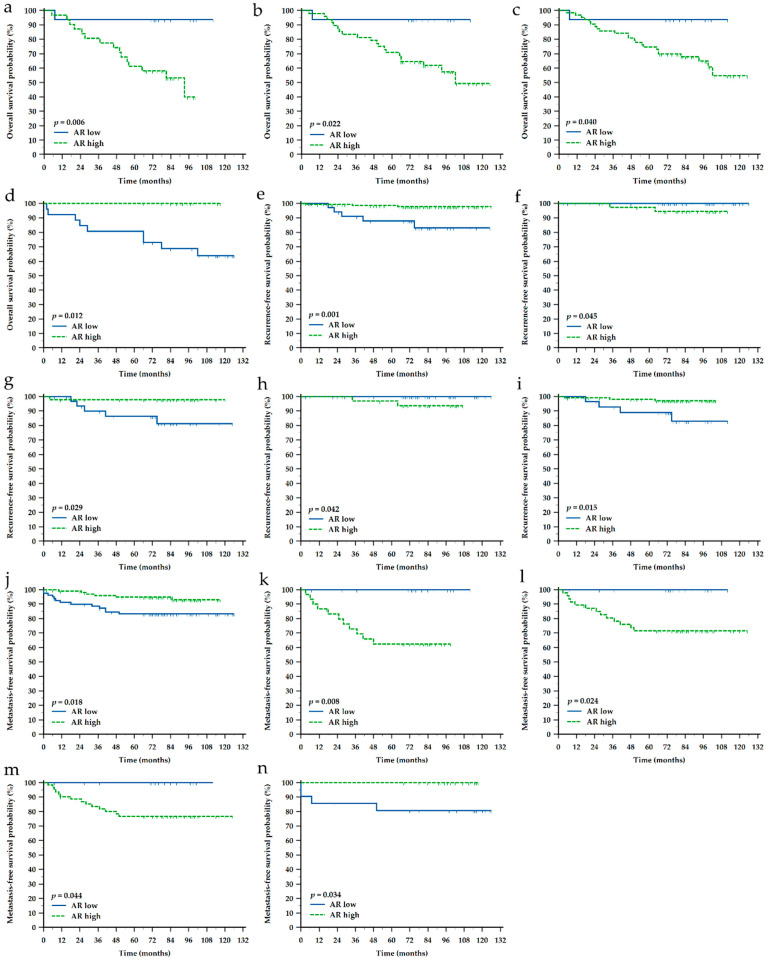
Kaplan-Meier survival curves depicting a significant impact of androgen receptor (AR) protein expression on different types of survival in different groups of breast cancer patients. (**a**) Impact on overall survival (OS) in TNBC patients (*n* = 47); (**b**) OS in ER-negative (*n* = 64); (**c**) OS in PR-negative (n = 79); (**d**) OS in HER2-positive (*n* = 64); (**e**) Impact on recurrence-free survival (RFS) in the whole cohort (N = 185); (**f**) RFS in ER-positive (*n* = 121); (**g**) RFS in PR-negative (*n* = 79); (**h**) RFS in PR-positive (*n* = 106); (**i**) RFS in HER2-negative (*n* = 142); (**j**) Impact on metastasis-free survival (MFS) in the whole cohort (N = 185); (**k**) MFS in TNBC (*n* = 47); (**l**) MFS in ER-negative (*n* = 64); (**m**) MFS in PR-negative (*n* = 79); (**n**) MFS in HER2-positive (*n* = 43). Shown are *p*-values for the long-rank test. Tick marks indicate censored cases.

**Table 1 life-12-01559-t001:** Baseline clinicopathological characteristics of Croatian cohort of breast cancer patients whose tumor tissue samples were used in this study.

Characteristic	*n* Out of 185 (%)
Age * (years)	60 (28–87)
Histotype	
Invasive ductal carcinoma	160 (86.5)
Invasive ductal carcinoma and ductal carcinoma in situ	25 (13.5)
Tumor size (T)	
T1 (≤2 cm)	105 (56.8)
T2 (2.1–5 cm)	71 (38.4)
T3 (>5.1 cm)	9 (4.9)
Nodal involvement (N)	
N0 (0)	113 (61.4)
N1 (1–3)	46 (25.0)
N2 (4–9)	14 (7.6)
N3 (≥10)	11 (6.0)
Distant metastasis (M)	
M0	183 (98.9)
M1	2 (1.1)
Ki-67	
<20%	53 (28.6)
≥20%	132 (71.4)
Lymphovascular invasion	
Absent	116 (62.7)
Present	69 (37.3)
Molecular subtype	
Luminal A	48 (25.9)
Luminal B	74 (40.0)
HER2-enriched	16 (8.6)
Triple-negative	47 (25.4)
Survival	
Alive	140 (75.7)
Deceased	45 (24.3)
Recurrence	
Absent	177 (95.7)
Present	8 (4.3)
Metastasis	
Absent	166 (89.7)
Present	19 (10.3)

* Data are presented as median (range).

**Table 2 life-12-01559-t002:** Association of SHH and AR protein expression with clinicopathological characteristics and receptor status of Croatian breast cancer patients. Statistically significant *p*-values are in bold.

Characteristic	SHH			AR		
	‘Low’	‘High’	*p*-Value	‘Low’	‘High’	*p*-Value
Age						
≤50 years	20 (43.5)	26 (56.5)	0.070	27 (58.7)	19 (41.3)	**0.017**
>50 years	40 (29.0)	98 (71.0)		53 (38.4)	85 (61.6)	
Histotype						
IDC	81 (50.6)	79 (49.4)	0.084	58 (36.2)	102 (63.7)	**0.017**
IDC + DCIS	8 (32.0)	17 (68.0)		3 (12.0)	22 (88.0)	
Tumor size						
T1	43 (41.0)	62 (59.0)	**0.026**	35 (33.3)	70 (66.7)	**0.002**
T2 + T3	46 (57.5)	34 (42.5)		45 (56.2)	35 (43.7)	
Nodal involvement						
0–3	124 (78.0)	35 (22.0)	0.052	17 (10.7)	142 (89.3)	0.087
≥4	15 (60.0)	10 (40.0)		0 (0.0)	25 (100.0)	
Ki-67						
<20%	23 (43.4)	30 (56.6)	0.418	8 (15.1)	45 (84.9)	**0.001**
≥20%	66 (50.0)	66 (50.0)		53 (40.2)	79 (59.8)	
LVI						
Absent	91 (78.4)	25 (21.6)	0.256	42 (36.2)	74 (63.8)	0.226
Present	49 (71.0)	20 (29.0)		19 (27.5)	50 (72.5)	
Molecular subtype *						
Luminal A	22 (45.8)	26 (54.2)	**0.037**	16 (33.3)	32 (66.7)	**<0.0001**
Luminal B	37 (50.0)	37 (50.0)		33 (44.6)	41 (55.4)	
HER2-E	5 (31.2)	11 (68.8)		12 (75.0)	4 (25.0)	
Triple-negative	32 (68.1)	15 (31.9)		40 (85.1)	7 (14.9)	
Receptor status						
ER-negative	31 (48.4)	33 (51.6)	**0.041**	43 (67.2)	21 (32.8)	**<0.0001**
ER-positive	40 (33.1)	81 (66.9)		10 (8.3)	111 (91.7)	
PR-negative	31 (39.2)	48 (60.8)	0.089	44 (55.7)	35 (44.3)	**<0.0001**
PR-positive	29 (27.4)	77 (72.6)		9 (8.5)	97 (91.5)	
HER2-negative	80 (56.3)	62 (43.7)	**0.028**	32 (22.5)	110 (77.5)	**0.008**
HER2-positive	16 (37.2)	27 (62.8)		2 (4.7)	41 (95.3)	
Survival						
Alive	101 (72.1)	39 (27.9)	0.209	73 (52.1)	67 (47.9)	0.239
Deceased	28 (62.2)	17 (37.8)		28 (62.2)	17 (37.8)	
Recurrence						
Absent	165 (93.2)	12 (6.8)	0.057	33 (18.6)	144 (81.4)	**0.003**
Present	6 (75.0)	2 (25.0)		5 (62.5)	3 (37.5)	
Metastasis						
Absent	77 (46.4)	89 (53.6)	0.167	67 (40.4)	99 (59.6)	**0.020**
Present	12 (63.2)	7 (36.8)		13 (68.4)	6 (31.6)	

* Protein expression was dichotomized using the median value of expression. AR—androgen receptor; DCIS—ductal carcinoma in situ; ER—estrogen receptor; HER2-E—HER2-enriched; IDS—invasive ductal carcinoma; LVI—lymphovascular invasion; PR—progesterone receptor; SHH—sonic hedgehog protein; T—tumor size; TNBC—triple-negative breast cancer.

**Table 3 life-12-01559-t003:** Correlation between SHH and AR protein expression in breast cancer patients classified by baseline clinicopathological characteristics and receptor status.

Characteristic	Category	ρ	*p*-Value
All patients		0.07	0.336
Age	≤50 years	0.14	0.356
	>50 years	0.02	0.787
Histotype	IDC	0.10	0.218
	IDC + DCIS	−0.13	0.533
Tumor size	<2 cm	−0.03	0.749
	≥2 cm	0.17	0.134
Nodal involvement	0–3	0.06	0.474
	≥4	0.13	0.534
Ki-67	<20%	0.18	0.205
	≥20%	0.04	0.615
Lymphovascular invasion	Absent	0.03	0.765
	Present	0.14	0.243
Molecular subtype	Luminal A	0.14	0.337
	Luminal B	−0.06	0.597
	HER2-E	0.11	0.680
	TNBC	−0.03	0.846
Receptor status	ER-negative	0.10	0.437
	ER-positive	0.02	0.847
	PR-negative	0.10	0.398
	PR-positive	0.04	0.717
	HER2-negative	0.10	0.247
	HER2-positive	−0.06	0.721
Survival	Alive	0.08	0.341
	Deceased	0.05	0.765
Recurrence	Absent	0.09	0.210
	Present	−0.55	0.157
Metastasis	Absent	0.09	0.246
	Present	0.03	0.900

ρ—Spearman’s rank correlation coefficient.

**Table 4 life-12-01559-t004:** Hazard ratios with 95% confidence intervals only for clinicopathological characteristics and sonic hedgehog protein and androgen receptor expressions with statistically significant (*p* < 0.05) impact on different types of survival in a cohort of Croatian breast cancer patients (N = 185). The *p*-values for the log-rank test are presented in Appendix A.

Characteristic			Molecular Subtype				Receptor Status					
		All	LumA	LumB	HER2-E	TNBC	ER-	ER+	PR-	PR+	HER2-	HER2+
Age ≤50 vs. >50 years	OS	2.5 (1.29–4.84)				3.7 (1.32–10.23)	3.2 (1.25–8.42)		2.6 (1.07–6.49)			N.A.
	RFS		N.A.									
T <2 vs. ≥2 cm	OS	2.7 (1.49–4.91)				3.1 (1.12–8.38)	2.6 (1.10–6.27)		2.9 (1.29–6.44)		3.6 (1.78–7.15)	
	MFS	5.0 (1.98–12.41)				3.6 (1.08–12.11)	4.3 (1.42–12.77)		4.3 (1.52–12.37)		5.7 (1.95–16.52)	
N 0–3 vs. ≥4	OS	4.5 (1.78–11.16)		7.0 (1.48–32.96)		7.0 (1.26–39.06)	4.4 (1.20–16.02)	4.2 (1.16–14.97)	5.1 (1.61–16.41)		14.7 (3.83–56.50)	
	MFS	19.9 (4.88–81.40)		11.2 (1.29–96.24)		19.0 (2.64–137.16)	9.6 (1.89–48.20)	47.2 (3.77–591.45)	10.20 (2.28–45.47)	24.9 (1.39–445.77)	133.9 (18.92–947.40)	
Ki-67 <20% vs. ≥20%	MFS	3.3 (1.25–8.81)									3.9 (1.35–10.99)	
LVI no vs. yes	OS		4.0 (1.07–15.11)									
SHH ‘low’ vs. ‘high’	OS				6.7 (1.06–42.82)		2.7 (1.03–6.96)					
	RFS	30.8 (1.63–580.74)				12.8 (1.40–116.76)			29.23 (1.56–547.50)		56.3 (2.46–1288.75)	
	MFS					8.2 (1.08–62.79)		0.19 (0.038–0.958)	5.4 (1.05–28.23)			
AR ‘low’ vs. ‘high’	OS					4.1 (1.48–11.23)	3.1 (1.17–8.20)		2.8 (1.05–7.47)			N.A.
	RFS	0.05 (0.009–0.323)						N.A.	0.16 (0.032–0.834)	N.A.	0.10 (0.016–0.639)	
	MFS	0.33 (0.133–0.824)				N.A.	N.A.		N.A.			N.A.

AR—androgen receptor; HER2-E—HER2-enriched; LumA—luminal A; LumB—luminal B; LVI—lymphovascular invasion; MFS—metastasis-free survival; N—number of positive lymph nodes; N.A.—no patients with certain characteristics have reach endpoint; OR—overall survival; RFS—recurrence-free survival; SHH—sonic hedgehog protein; T—tumor size; vs.—versus.

## Data Availability

Individual per patient data are available from the corresponding authors upon reasonable request.

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
