# Peer review of "Higher Expressions of SHH and AR Are Associated with a Positive Receptor Status and Have Impact on Survival in a Cohort of Croatian Breast Cancer Patients"

_life, 2022, doi:10.3390/life12101559_

Round 1

Reviewer 1 Report

In this manuscript, the author immunohistochemically determined expression of androgen receptor (AR) and sonic hedgehog protein (SHH), the main ligand of the Hedgehog-GLI signaling pathway, in 185 archival primary BC tissue samples and correlated it with clinicopathological characteristics, molecular sub types, receptors’ statuses, and survival in a cohort of Croatian BC patients. And finally clarified interaction between Hedgehog-GLI pathway and studied receptors could be a valuable therapeutic target. This is a very interesting study. With editing and some minor revisions, I feel that this manuscript will be suitable for publication.

1.      Figure 1 did not clearly show the location of the different percentages of AR-positive tumor nuclei.

2.      Figure 2 did not clearly show the location of the different levels of SHH protein expression.

3.      This study is to illustrate a major finding, the sample size is too small. Could the authors provide more samples to verify the point of view?

4.      In the manuscript, p-values do not appear in the main text after each data, the authors may be able to place specific data in the Table or Supplementary material.

5.      Line 357-359, In our study SHH was expressed in vast majority of BC tissue samples. We found a negative correlation between SHH expression and tumor size, what is opposite from previous studies.” However, the explanation that follows does not clearly clarify why the results in the paper are different from those reported previously.

6.      The grammar of this manuscript should be revised.

Author Response

In this manuscript, the author immunohistochemically determined expression of androgen receptor (AR) and sonic hedgehog protein (SHH), the main ligand of the Hedgehog-GLI signaling pathway, in 185 archival primary BC tissue samples and correlated it with clinicopathological characteristics, molecular sub types, receptors’ statuses, and survival in a cohort of Croatian BC patients. And finally clarified interaction between Hedgehog-GLI pathway and studied receptors could be a valuable therapeutic target. This is a very interesting study. With editing and some minor revisions, I feel that this manuscript will be suitable for publication.

We thank the reviewer for carefully reading our manuscript and suggesting valuable corrections and further improvements.

  1.      Figure 1 did not clearly show the location of the different percentages of AR-positive tumor nuclei.

The representative IHC photomicrographs were meticulously selected by experienced pathologist, however, we are aware that Figure 1 in present form is maybe too small to precisely show the location of AR-positive tumor nuclei. For that reason, we are providing original photomicrographs from which the Figure 1 was assembled as the Supplemental Figure S1.

  1.      Figure 2 did not clearly show the location of the different levels of SHH protein expression.

As stated previously, we are aware that also Figure 2 in present form is maybe too small to precisely show the location of the different levels of SHH protein expression. For that reason, we are providing also original photomicrographs from which the Figure 2 was assembled as the Supplemental Figure S2.

  1.      This study is to illustrate a major finding, the sample size is too small. Could the authors provide more samples to verify the point of view?

Unfortunately, at this moment we cannot provide more samples. However, the sample size of our study is in essence not arbitrary, but it has been selected by performing a power analysis based on the results obtained by Riaz et al. Cell Commun Signal. 2018;16(1):3.

  1.      In the manuscript, p-values do not appear in the main text after each data, the authors may be able to place specific data in the Table or Supplementary material.

To be completely honest we do not fully understand this suggestion. We agree that explaining the results of statistical analyses in words should generally be supported by providing also actual p-values in the text. However, since we have obtained so many p-values, in the text we have stated only those which we think are the most important to support the credibility of descriptive explanation of statistical results. Whatsoever, all p-values we obtained are presented in both main and supplementary tables and repeating all of them in the main text we believe would be redundant and could make following the text more difficult.

  1.      Line 357-359, “In our study SHH was expressed in vast majority of BC tissue samples. We found a negative correlation between SHH expression and tumor size, what is opposite from previous studies.” However, the explanation that follows does not clearly clarify why the results in the paper are different from those reported previously.

The most significant associations of tumor size with other clinicopathological characteristics and studied protein expressions we observed when we divided tumor samples into ≤2 cm (T1) and >2.1 cm (T2 and T3) groups. However, if leave all three T categories and binarize SHH expression according to its median value, the highest SHH expression is observed for T3, then for T1, and lowest for T2 group (data not presented). But since we had only 9 T3 samples, we decided that merging T2 and T3 groups would give statistically more reliable results. In addition, another possible explanation of discrepancy between ours and previous results is that studies which have shown a positive correlation between SHH expression and tumor size primarily studied SHH mRNA expression. We extended our explanation in the text.

  1.      The grammar of this manuscript should be revised.

The text has been thoroughly inspected and all observed grammar and spelling errors were corrected.

Reviewer 2 Report

This study is very interesting. The authors add new insights into the roles of the HH-GLI pathway and AR in BC. They conclude that AR may have a protective role in receptor positive BC patients. In fact, they observe that high AR expression has a positive impact on survival, with fewer relapses of local and distant diseases. Therefore, higher AR expression showed to be a marker for shorter OS in receptor negative and TNBCs. However, no correlation was observed between AR and SHH expressions in any of the BC patient subgroups. The importance of the HH-GLI pathway is highlighted by its role in breast cancer invasiveness, lymph node metastases and relapses. The authors show that SHH, the ligand of this pathway, has a negative impact negative impact on a survival of sex hormone receptor-negative and TNBC patients, while the opposite is observed for HER2-E patients. However, in line 398, they conclude that the HH-GLI pathway is harmful for TNBC and HER-E BC patients. This is in contradiction with the previous sentence (line 391). Could the authors review it? Before the publication the article small comments are needed:

- in table 1, could the authors add the number of patients enrolled in this study, in the row (for example, n 185 (%) of the table?

- the authors on lines 367-368 state that NF-kB is a possible regulator of SHH expression in BC, it could be interesting to study the relationship between ER activation and SHH expression, considering the positive crosstalk between the receptor of estrogen and NFκB in breast cancer (Frasor J, Weaver A, Pradhan M, et al. Positive cross-talk between estrogen receptor and NF-kappaB in breast cancer [published correction appears in Cancer Res. 2010 Jan 15;70(2):854] Frasor J, El-Shennawy L, Stender JD, Kastrati I. NFκB affects estrogen receptor expression and activity in breast cancer through multiple mechanisms. Mol Cell Endocrinol. 2015 Dec 15;418 Pt 3(0 3):235-9). 

- it might be interesting for the authors to evaluate the hormone level in patients with BC, in order to correlate these hormone levels with the observed effects.

- small revisions to the text (see attached file). 

Author Response

This study is very interesting. The authors add new insights into the roles of the HH-GLI pathway and AR in BC. They conclude that AR may have a protective role in receptor positive BC patients. In fact, they observe that high AR expression has a positive impact on survival, with fewer relapses of local and distant diseases. Therefore, higher AR expression showed to be a marker for shorter OS in receptor negative and TNBCs. However, no correlation was observed between AR and SHH expressions in any of the BC patient subgroups. The importance of the HH-GLI pathway is highlighted by its role in breast cancer invasiveness, lymph node metastases and relapses. The authors show that SHH, the ligand of this pathway, has a negative impact negative impact on a survival of sex hormone receptor-negative and TNBC patients, while the opposite is observed for HER2-E patients. However, in line 398, they conclude that the HH-GLI pathway is harmful for TNBC and HER-E BC patients. This is in contradiction with the previous sentence (line 391). Could the authors review it? Before the publication the article small comments are needed:

We thank the reviewer for carefully reading our manuscript and suggesting valuable corrections and further improvements. However, we disagree that statements in lines 391 and 398 are contradictory, except that one in line 391 probably needs more detailed explanation. We observed that although higher SHH expression was associated with sex hormone receptors-positive status, its negative impact on survival was observed in sex hormone receptor-negative and TNBC patients. To the contrary, both higher SHH expression and its negative impact on survival were observed in HER2-E BC patients. We added that more precise explanation to the statement in line 391.

- in table 1, could the authors add the number of patients enrolled in this study, in the row (for example, n 185 (%) of the table?

Thank you for this suggestion, however, in the Table 1 we rather put ‘n out of 185 (%)’ because just ‘n 185 (%)’ could seem incomprehensive.

- the authors on lines 367-368 state that NF-kB is a possible regulator of SHH expression in BC, it could be interesting to study the relationship between ER activation and SHH expression, considering the positive crosstalk between the receptor of estrogen and NFκB in breast cancer (Frasor J, Weaver A, Pradhan M, et al. Positive cross-talk between estrogen receptor and NF-kappaB in breast cancer [published correction appears in Cancer Res. 2010 Jan 15;70(2):854] Frasor J, El-Shennawy L, Stender JD, Kastrati I. NFκB affects estrogen receptor expression and activity in breast cancer through multiple mechanisms. Mol Cell Endocrinol. 2015 Dec 15;418 Pt 3(0 3):235-9).

We agree with the reviewer and thank for this suggestion. We would definitively take that into consideration for our following studies in which we plan to inspect interactions between ER and SHH more mechanistically.

- it might be interesting for the authors to evaluate the hormone level in patients with BC, in order to correlate these hormone levels with the observed effects.

We agree that evaluating both cholesterol and hormone levels in BC patients could be very interesting information in the light of the fact that activation of Hedgehog-GLI signaling stimulates the conversion of cholesterol to steroids (Tang et al. Cell Signal. 2015;27(3):487-97). However, since this is a retrospective study, doing that retroactively would be neither feasible (45 of our patients are already deceased) nor meaningful (present hormone levels would not reflect their status in the time of disease diagnosis).

- small revisions to the text (see attached file).

We thank again the reviewer for carefully reading our manuscript and suggesting valuable corrections. We have implemented them considering general grammar rules and instructions for authors of the Life journal.